# Bacterial fumarase and L-malic acid are evolutionary ancient components of the DNA damage response

Esti Singer[1], Yardena BH Silas[1,2], Sigal Ben-Yehuda[1], Ophry Pines[1,2]*

[1]Department of Microbiology and Molecular Genetics, IMRIC, Faculty of Medicine, Hebrew University, Jerusalem, Israel; [2]CREATE-NUS-HUJ Program and the Department of Microbiology, Yong Loo Lin School of Medicine, National University of Singapore, Sinapore

**Abstract** Fumarase is distributed between two compartments of the eukaryotic cell. The enzyme catalyses the reversible conversion of fumaric to L-malic acid in mitochondria as part of the tricarboxylic acid (TCA) cycle, and in the cytosol/nucleus as part of the DNA damage response (DDR). Here, we show that fumarase of the model prokaryote *Bacillus subtilis* (Fum-bc) is induced upon DNA damage, co-localized with the bacterial DNA and is required for the DDR. Fum-bc can substitute for both eukaryotic functions in yeast. Furthermore, we found that the fumarase-dependent intracellular signaling of the *B. subtilis* DDR is achieved via production of L-malic acid, which affects the translation of RecN, the first protein recruited to DNA damage sites. This study provides a different evolutionary scenario in which the dual function of the ancient prokaryotic fumarase, led to its subsequent distribution into different cellular compartments in eukaryotes.
DOI: https://doi.org/10.7554/eLife.30927.001

## Introduction

The enzyme fumarase (classII, fumarate hydratase in higher eukaryotes) is a conserved protein in all organisms from bacteria to human with respect to its sequence, structure, and enzymatic activity (*Akiba et al., 1984*). Fumarase is a dual targeted protein in eukaryotes; its echoforms are distributed between mitochondria and the cytosol/nucleus (*Yogev et al., 2011*; *Yogev et al., 2010*). Moonlighting of proteins is also a well-known phenomenon, which is defined by single proteins that can perform different functions in the cell (*Gancedo et al., 2016*; *Espinosa-Cantú et al., 2015*). Fumarase is also a moonlighting protein since it performs functions in the tricarboxylic acid (TCA) cycle in mitochondria and it participates in the DNA damage response (DDR) in the nucleus (*Boukouris et al., 2016*). In the TCA cycle, fumarase converts fumaric acid to L-malic acid while in the nucleus it catalyses the opposite reaction, thereby supplying fumaric acid as a signaling molecule for the DDR (*Yogev et al., 2010*). In human cells, fumaric acid has been shown to inhibit certain histone dimethylases and prolyl hydroxylases (*Jiang et al., 2015*; *Gottlieb and Tomlinson, 2005*; *Isaacs et al., 2005*). Worth mentioning is that fumarase associates with different protein partners in the two compartments (in the mitochondria where it is part of the TCA cycle, it interacts with other TCA cycle enzymes such as malate dehydrogenase, while in the cytosol/nucleus, it interacts with components of the DDR such as kinases and histones). Two questions with regard to the evolution of fumarase come to mind; (1) how and when did dual targeting of the protein evolve? And (2) how and when did dual function of the protein evolve?

With regard to the first question, fumarase has been shown to distribute between the mitochondria and the cytosol/nucleus by different mechanisms in different eukaryotes (*Yogev et al., 2011*). In Arabidopsis, there are two fumarase-encoding genes, which are highly homologous, besides the fact

*For correspondence:
ophryp@ekmd.huji.ac.il

Competing interests: The authors declare that no competing interests exist.

**eLife digest** Living cells make an enzyme called fumarase. It converts a chemical called fumaric acid into L-malic acid. This is a crucial step in primary metabolism and aerobic respiration, the process of using oxygen to release energy for life. Yet it is not the only role that fumarase plays. In the cells of eukaryotes such as plants, animals and even baker's yeast, aerobic respiration happens inside compartments called mitochondria. Yet fumarase is also found in the nucleus, which contains the cell's genetic material. Inside the nucleus, this enzyme takes part in the DNA damage response that senses and repairs damage to the genetic code.

Simpler organisms, like bacteria, do not have mitochondria or a nucleus. Instead, all their reactions take place inside the main space within the cell. The current model for the evolution of fumarase is that the enzyme evolved in an ancient bacterium for the production of energy. Then, in more complex organisms, becoming split between the mitochondria and the nucleus allowed it to take on a second role in the DNA damage response. Singer et al. now challenge that model, and show that fumarase takes part in DNA damage repair in bacteria too.

*Bacillus subtilis* has one fumarase gene, known as *fum-bc*. Singer et al. showed that, without this gene, the bacteria do not grow well under conditions where they need to use aerobic respiration. But, the bacteria also became sensitive to DNA-damaging agents such as ionizing radiation or a chemical called methyl methanesulfonate.

Singer et al. then expressed the bacterial *fum-bc* gene in baker's yeast, *Saccharomyces cerevisiae*. This organism has mitochondria and a cell nucleus. With the yeast's own fumarase gene switched off, the bacterial fumarase was able to take on both roles – aerobic respiration and the DNA damage response.

In bacteria grown with the DNA-damaging chemical, the level of fumarase started to rise. A fluorescent tag revealed that it also changed location, moving close to the bacteria's DNA. As such, even in bacteria, fumarase has two roles. Further experiments showed that the L-malic acid made by fumarase affects the production of a protein called RecN, and it is this protein that triggers DNA repair.

These findings shed new light on the evolution of fumarase, and suggest that its dual role evolved before its dual location in eukaryotes. The next step is to find out exactly how L-malic acid affects the production of RecN.

DOI: https://doi.org/10.7554/eLife.30927.002

that one encodes a mitochondrial targeting signal (MTS), while the other lacks it (*Pracharoenwattana et al., 2010*). In human, there is a single gene, however, it makes multiple mRNAs, which either encode or lack a MTS. In these cases, described above, two different types of mRNA are made which determine dual localization of fumarase echoforms (*Dik et al., 2016*). In yeast (*S. cerevisiae*), mitochondrial and cytosolic echoforms of fumarase, are encoded by a single nuclear gene (FUM1) and follow an intriguing mechanism of protein subcellular localization and distribution (*Yogev et al., 2011*). Translation of all FUM1 messages initiates only from the 5′-proximal AUG codon and results in a single translation product that contains the MTS (*Yogev et al., 2007*). The precursor of yeast fumarase is first partially translocated into mitochondria, so that the N-terminal signal is cleaved. While a subset of these molecules continues to be fully translocated into the organelle, the rest are folded into an import-incompetent conformation and are released by the retrograde movement back into the cytosol. Thus, protein folding is the driving force for fumarase dual targeting in yeast, a mechanism termed reverse translocation (*Karniely and Pines, 2005*; *Kalderon and Pines, 2014*). Dual localization is a very abundant phenomenon in eukaryotes, and in fact, we estimate that a third of the yeast mitochondrial proteome is dual targeted (*Ben-Menachem et al., 2011*). Therefore, dual targeting or dual localization of proteins is a major outcome of gene expression and as such, the evolutionary pressures governing this phenomenon are of fundamental importance. In a major discovery, we have shown that dual-targeted proteins are significantly more evolutionary conserved than exclusive mitochondrial proteins. We reached this conclusion by employing codon usage bias, propensity for gene loss, phylogenetic relationships, conservation analysis at the DNA level, and gene expression (*Dinur-Mills et al., 2008*; *Kisslov et al.,*

*2014*). This has changed the way we think about dual targeting since we now assume that the majority of dual targeted proteins have discrete functions in the different subcellular compartments, regardless of their dual-targeting mechanism. Thus we hypothesize that dual targeting is maintained due to separate selective pressures administered by the different compartments to maintain the functions associated with the protein sequences (*Kisslov et al., 2014*). Here, we use the enzyme fumarase as a paradigm of this evolution and show for the first time that the single fumarase gene of *B. subtilis* is involved in the bacterial DDR, in addition to its role in the TCA cycle. This finding suggests that dual function of fumarase, in the bacterial progenitor, preceded the dual targeting that we find in eukaryotes. Intriguingly, fumarase in bacteria, similarly to eukaryotes, appears to be linked to the DDR by metabolite signaling, however the active molecule in this case is L-malic acid and not fumaric acid.

## Results

### Bacterial fum-bc functions in the TCA cycle and the DDR

The conservation of fumarase dual function in eukaryotes without conservation of the mechanism of dual targeting has brought us to examine the following hypothesis: Dual function of fumarase in bacteria preceded dual targeting in eukaryotes; in other words, it occurred in the prokaryotic ancestor prior to evolving of eukaryotes. Thus, dual function was the driving force for fumarase dual localization in the eukaryotic cell. To examine this question we employed the Gram positive bacterium *Bacillus subtilis,* which contains a single class II fumarase gene (*fum-bc*). As shown in *Figure 1A* (compare the right and left panels), *B. subtilis* deleted for the *citG* gene (*fum-bc*) which encodes fumarase, exhibits very poor growth on defined medium (S7) containing a low level of glucose (0.1%) that requires respiration and the TCA cycle for efficient growth. Thus, as expected, Fum-bc has a crucial function within the TCA cycle. Intriguingly, *fum-bc* deleted strains are sensitive to DNA damaging agents such as ionized radiation (IR) (*Figure 1A*, middle panel) or methyl methanesulfonate (MMS) (*Figure 1B*). We expressed Fum-bc from the ectopic amyE locus (Δfum+Fum-bc, *Figure 1—figure supplement 1*) and this strain exhibits similar resistance to MMS treatment as the wild type (*Figure 1B*, compare the first and third rows). The slight difference in resistance may be since fumarase is expressed from the amyE locus and not from its native promoter. Quantitative experiments following colony forming units (CFU) are shown in *Figure 1Ci* and 1Cii. The difference between wild type and Δfum strains is insignificant (compare the two left bars of *Figure 1Ci* and *Figure 1Cii*) while the difference between these strains following DNA damage is highly significant (bars 4 and 5, p<0.01). These results suggest, as we hypothesized, that dual function of fumarase can already be found in prokaryotes.

To examine the ability of Fum-bc to perform its two functions (TCA and DDR) in a eukaryotic model, we took advantage of our experience with the yeast *Saccharomyces cerevisiae*. *fum-bc* was cloned into a yeast expression vector under the control of the *GAL* promoter. The levels of fumarase in cultures grown in galactose medium of wild type yeast, yeast deleted for the chromosomal *FUM1* and such a deletion strain expressing bacterial *fum-bc*, are shown in *Figure 1—figure supplement 2*. The *fum-bc* gene was expressed in a yeast strain deleted for the endogenous *FUM1* gene (Δ*fum1*). This strain was grown on glucose (dextrose) as a control medium that does not require respiration for growth, and on ethanol which does require respiration and the function of the TCA cycle. As shown in *Figure 1D*, yeast strains deleted for the endogenous *FUM1* and expressing *fum-bc* can partially complement the lack of yeast fumarase for growth on ethanol (compare row 4 [fum-bc expression] to row 2 [no fumarase expression]). Fum-bc does not contain a mitochondrial targeting sequence and it is not targeted to, or imported into mitochondria. We have performed subcellular fractionation of yeast cells expressing fum-bc. The bacterial protein is located only in the 'cytosol' and not in mitochondria (*Figure 1—figure supplement 3*). From previous studies, it turns out that fumarase, located outside mitochondria, can nevertheless function in the TCA cycle (*Sass et al., 2003*; *Stein et al., 1994*). The explanation for this is that the metabolites fumarate and malate can enter and exit the organelle (via specific inner membrane transporters) thereby completing the TCA cycle.

To examine the activity of Fum-bc in the DNA damage response, the yeast FUM^m strain was employed. The FUM^m strain harbors a chromosomal FUM1 deletion (Δ*fum1*) and a FUM1 ORF (open

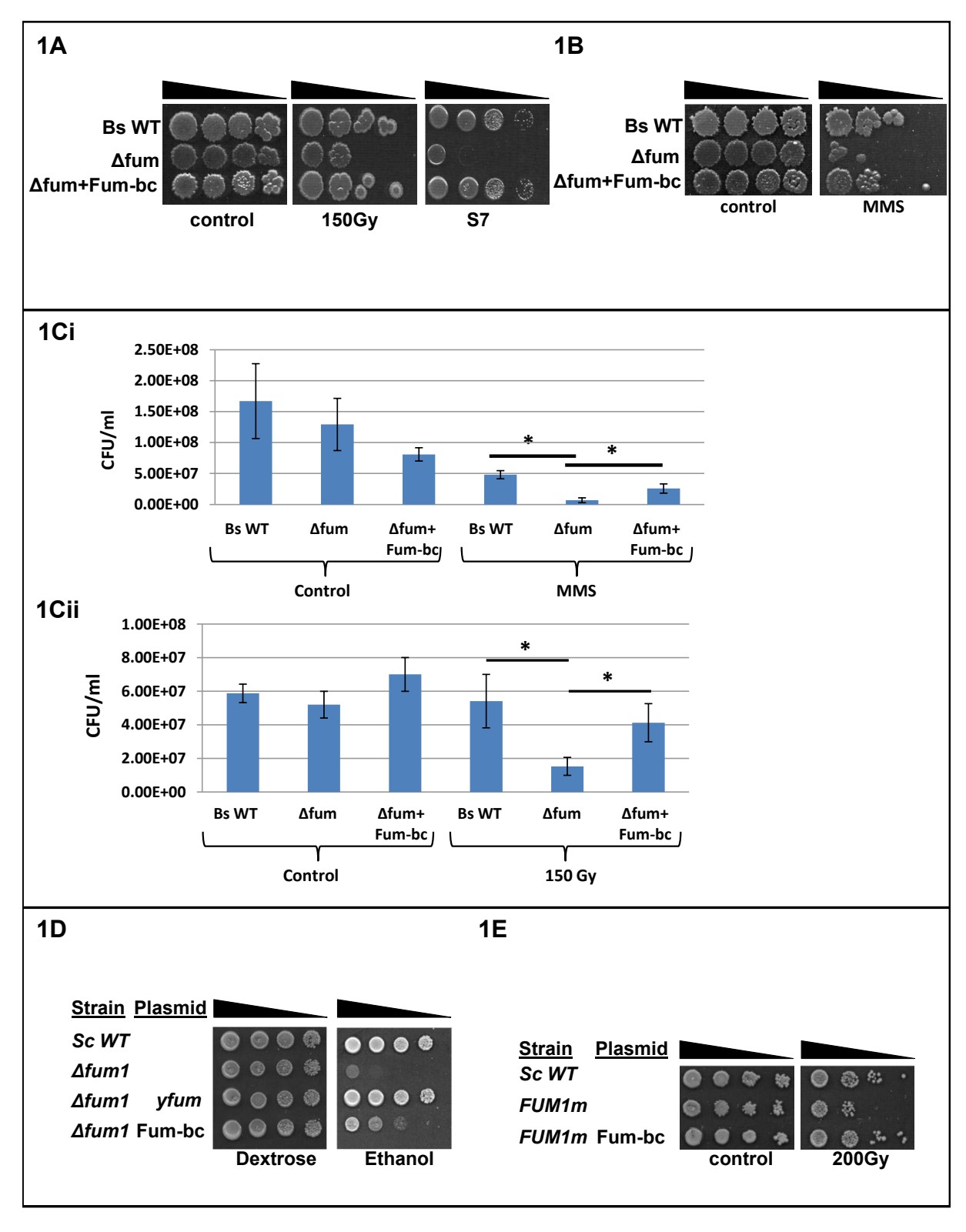

**Figure 1.** Fumarase is required for an efficient DNA damage response in *Bacillus subtilis*. (**A**) *Bacillus subtilis* wild type (Bs WT) and Δfum (=ΔcitG) strains were untreated (control) or exposed to ionized radiation (150 Gy) and then serially diluted onto LB plates or minimal S7 medium plates (containing 0.1% glucose). (**B**) *Bacillus subtilis* Δfum strain is sensitive to DNA damage. *Bacillus subtilis* wild type (Bs WT), Δfum and Δfum+fum-bc strains were grown to logarithmic phase and MMS (methyl methanesulfonate) was added to a final concentration of 0.07%(v/v) for 45 min. The cells were then
*Figure 1 continued on next page*

Figure 1 continued

washed and serially diluted onto LB plates. (C) Quantitative analysis of *Bacillus subtilis* Δfum *strain* sensitivity to DNA damage. (Ci) MMS: *Bacillus subtilis* wild type (Bs WT), Δfum and Δfum+fum-bc strains were grown to logarithmic phase and MMS (methyl methanesulfonate) was added to a final concentration of 0.07%(v/v) for 50 min. Colony forming units (CFU) were determined after plating serial dilutions of cell cultures on LB agar plates. (mean ±SEM [n = 3], p=0.01). (Cii) Ionized radiation (IR): *Bacillus subtilis* wild type (Bs WT) and Δfum strains, grown as above, were untreated (control) or exposed to IR (150 Gy). The CFU was determined as above. (mean ±SEM [n = 3], p=0.04). (D) fum-bc can complement DNA damage sensitivity in yeast. *Saccharomyces cerevisiae* wild type (Sc WT) and Δ*fum1* strains harboring the indicated plasmids were serially diluted and grown on glucose (dextrose) or ethanol SD medium plates. (E) fum-bc can complement respiration in yeast. *Saccharomyces cerevisiae* wild type (Sc WT), *FUM1*$^m$ and *FUM1*$^m$ transformed with a plasmid-encoding *Bacillus subtilis* fumarase, were grown to logarithmic phase in galactose SD medium, irradiated and serially diluted onto galactose SD plates. The data presented in *Figure 1*, in each case, represent the results of three similar experiments.

DOI: https://doi.org/10.7554/eLife.30927.003

The following figure supplements are available for figure 1:

**Figure supplement 1.** Expression of B. subtilis and yeast fumarases.
DOI: https://doi.org/10.7554/eLife.30927.004
**Figure supplement 2.** Expression of B. subtilis and yeast fumarases.
DOI: https://doi.org/10.7554/eLife.30927.005
**Figure supplement 3.** Subcellular fractionation of yeast cells.
DOI: https://doi.org/10.7554/eLife.30927.006

reading frame) insertion in the mitochondrial DNA, thereby allowing exclusive mitochondrial fumarase (Fum1) expression, but lacking extra-mitochondrial (cytosolic/nuclear) fumarase (*Yogev et al., 2010*). Accordingly, the FUM$^m$ strain exhibits a functional TCA cycle and the ability to respire; however, it displays sensitivity to DNA damaging agents such as ionized radiation. When *fum-bc* is expressed in the yeast FUM$^m$ strain, it can complement the lack of extra-mitochondrial fumarase with respect to sensitivity to ionized radiation (*Figure 1E*, compare row 3 [cytosolic fum-bc expression] to row 2 [no cytosolic fumarase expression]). These results support our hypothesis that bacterial fumarase has the capacity to function both in the TCA cycle and the DNA damage response.

## Fum-bc is induced upon DNA damage, and is co-localized with the bacterial DNA

To examine the role of fumarase in the DDR, *Bacillus subtilis* cells were grown to log phase and then incubated in the presence or absence of MMS. Cell extracts were analyzed with time by western blot using a mixture of anti yeast fumarase (anti yFum) and anti human fumarase-FH (anti hFum) antibodies. As shown in *Figure 2A* the level of Fum-bc gradually increases (compare the top panel to the SigmaA loading control in the lower panel. After 60 min of MMS treatment, the amount of fumarase rises two fold (*Figure 2B*). This finding is consistent with the proposed function of Fum-bc in the DDR. As shown In *Figure 2—figure supplement 1*, the levels of other TCA cycle enzymes, citrate synthase 2 (citZ) and isocitrate dehydrogenase (ICDH), do not change significantly upon treatment with MMS.We next asked whether the appearance and localization of Fum-bc changes upon induction of the DNA damage response. For this, we monitored fluorescence in *B. subtilis* strains harboring genomic *Fum-bc-GFP* fusions, in combination with DAPI for DNA staining and FM4-64 for membrane staining. *Fum-bc-GFP* retains full fumarase activity in cell extracts (*Figure 2—figure supplement 2*). As shown in *Figure 2C*, in untreated cells (top panels) Fum-bc does not generally colocalize with the bacterial DNA (top right panel); only 13% of *Fum-bc-GFP* foci showed colocalization with the DNA (*Figure 2D*). Following treatment with MMS, we detected full colocalization of *Fum-bc-GFP* fluorescence with the DNA DAPI stain (bottom right panel); more than 95% of the foci showed colocalization with the DNA (*Figure 2D*). Upon DNA damaging treatment, fumarase clearly coincides with the condensed DNA and in fact as shown in the two lower right panels of *Figure 2C*, the *Fum-bc-GFP* fluorescence and the DNA DAPI stain perfectly coincide during this process. Without DNA damaging treatment the fum-bc-GFP foci and DAPI stained DNA do not regularly superimpose. Nevertheless one does see some co-staining which can be explained by simple coincidence or naturally occurring low random DNA damage for example at DNA replication sites. Furthermore, we found that without induction of double strand breaks (DSBs) (-MMS) most of the cells show between three to four *Fum-bc-GFP* foci, while after induction of DSB (+MMS) most of the cells show one to two extensive foci that overlap with the DAPI stained DNA (*Figure 2E*). Thus, Fum-bc appears to be

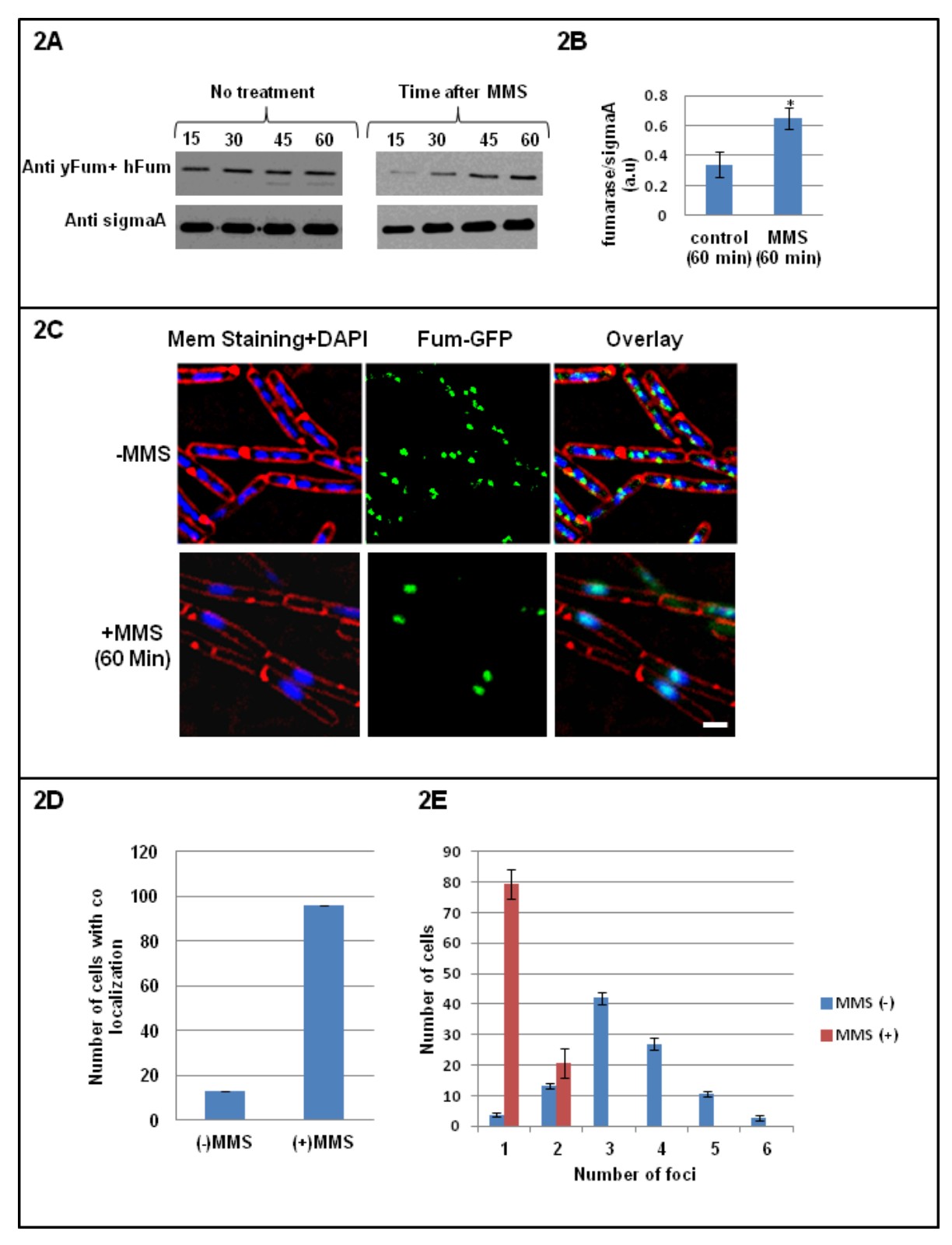

**Figure 2.** Fumarase is co-localized with the DNA after induction of DNA damage. (**A**) *B. subtilis* was grown to logarithmic phase, MMS was added to final concentration of 0.07%(v/v) for 60 min. The levels of fumarase in the cells were determined at the indicated times by western blot analysis. Anti-SigmaA was used as a loading control. (**B**) The chart presents the relative amount of fumarase (densitometric analysis of *Figure 2A*, normalized to SigmaA) in cells grown in the absence or presence of MMS at 60 min (mean ±SEM [n = 5], p=0.01). (**C**) Fluorescence microscopy images of *B. subtilis*

*Figure 2 continued on next page*

*Figure 2 continued*

expressing fum-GFP in the absence (upper panel) and presence (lower panel) of MMS. FM4-64 (red), DAPI (blue), GFP (green) and a merge are shown. One representative experiment out of three independent experiments is shown. Bar, 1 μm. (D) Colocalization of Fum-bc-GFP and DAPI staining. The number of *B. subtilis* cells showing colocalization between GFP and DAPI staining with or without DNA damage induction by MMS. Error bars represent standard deviation (SD) of the mean fluorescence signal calculated from at least 100 cells. (E) The number of Fum-bc-GFP foci per cell with or without MMS. Shown is a representative experiment out of three independent repeats. (mean ±SEM [n = 3]).

DOI: https://doi.org/10.7554/eLife.30927.007

The following figure supplements are available for figure 2:

**Figure supplement 1.** TCA cycle enzymes.

DOI: https://doi.org/10.7554/eLife.30927.008

**Figure supplement 2.** Enzymatic activity of *B. subtilis* fumarase fused to GFP.

DOI: https://doi.org/10.7554/eLife.30927.009

recruited to the DNA during the DNA damage response supporting a role for the bacterial fumarase in this response.

## Fumarase activity and the product of its reaction, L-malic acid, are required for the DDR

To examine whether the role of Fum-bc in the DDR requires its enzymatic activity, we first identified mutations within the fumarase active site that may abolish its enzymatic activity. According to the literature (**Weaver et al., 1997**; **Alam et al., 2005**; **Kokko et al., 2006**) fumarase has a known active site in eukaryotes (*S. cerevisiae*) and in prokaryotes (*Escherichia coli*). Based on sequence similarity between eukaryotic and prokaryotic fumarases, we created two separate substitution mutations within the suspected Fum-bc active site; H186N and H127R. H186N has been suggested to be a residue of the active site of *E. coli* fumarase while H127R has been suggested to be a residue of the active site of the eukaryotic *S. cerevisiae* fumarase. We created *fum-bc* point mutations corresponding to the above single amino acid changes in the respective active sites, and examined expression of the proteins by western blot (**Figure 3—figure supplement 1**). While extracts of cells expressing only the H186N mutated fumarase, were essentially devoid of enzymatic activity, those expressing H127R displayed 40% of the wild type activity (**Figure 3—figure supplement 2**). Accordingly, cells expressing H186N exhibit sensitivity to MMS and defective growth on S7 medium (low glucose, **Figure 3A**, right panel), while H127R grows normally on S7 plates and is not sensitive to MMS (**Figure 3—figure supplement 3**). This indicates that enzymatic activity is required for both DDR and respiration-related functions of fumarase.

Fumarase catalyses the reversible conversion of fumaric acid to L-malic acid as part of the TCA cycle in mitochondria. In yeast and human cells, defective for extra mitochondrial fumarase, the sensitivity to DNA damage can be complemented by fumaric acid, added to the growth medium in the form of an ester (monoethyl fumarate, which is cleaved in the cells to form the free acid) (**Yogev et al., 2010**; **Jiang et al., 2015**). To examine if products or substrates of the fumarase reaction in *B. subtilis* may complement the lack of fumarase in the DDR, bacteria were grown in the presence of organic acids added to the medium. As shown in **Figure 3B**, *B. subtilis* cells deleted for the *fum-bc* gene are protected from the DNA damaging treatment with MMS, by L-malic acid (compare the third and fourth rows of the right and left panels). In contrast, succinic and citric acids have no protective effect (two middle panels respectively). Since, fumaric acid is not soluble; we also examined the capacity of esters of the other organic acids - to protect the bacterial cells against MMS. While diethylmalate protects the cells from MMS the other organic acids (monoethyl fumarate and monoethyl succinate), had a much weaker effect (**Figure 3—figure supplement 4**).

Organic acids, and in particular L-malic acid, appear to play a role in the DNA damage response in *B. subtilis*. To correlate changes in the intracellular levels of these organic acids we employed GC-MS of cell lysates. As shown in **Figure 3C**, following induction of DNA damage with MMS, the relative levels of L-malic acid increase while those of succinic and fumaric acids decrease (compare the two left sets of bars). This clearly fits the role of L-malic as a DDR signaling molecule and that succinic and fumaric acids do not have such a role. In *B. subtilis*, strains deleted for *fum-bc*, the single fumarase gene, accumulate higher levels of fumaric and succinic acid (**Figure 3C**). This is expected from a block in the TCA cycle at the conversion step of succinic to fumaric acid and subsequently to

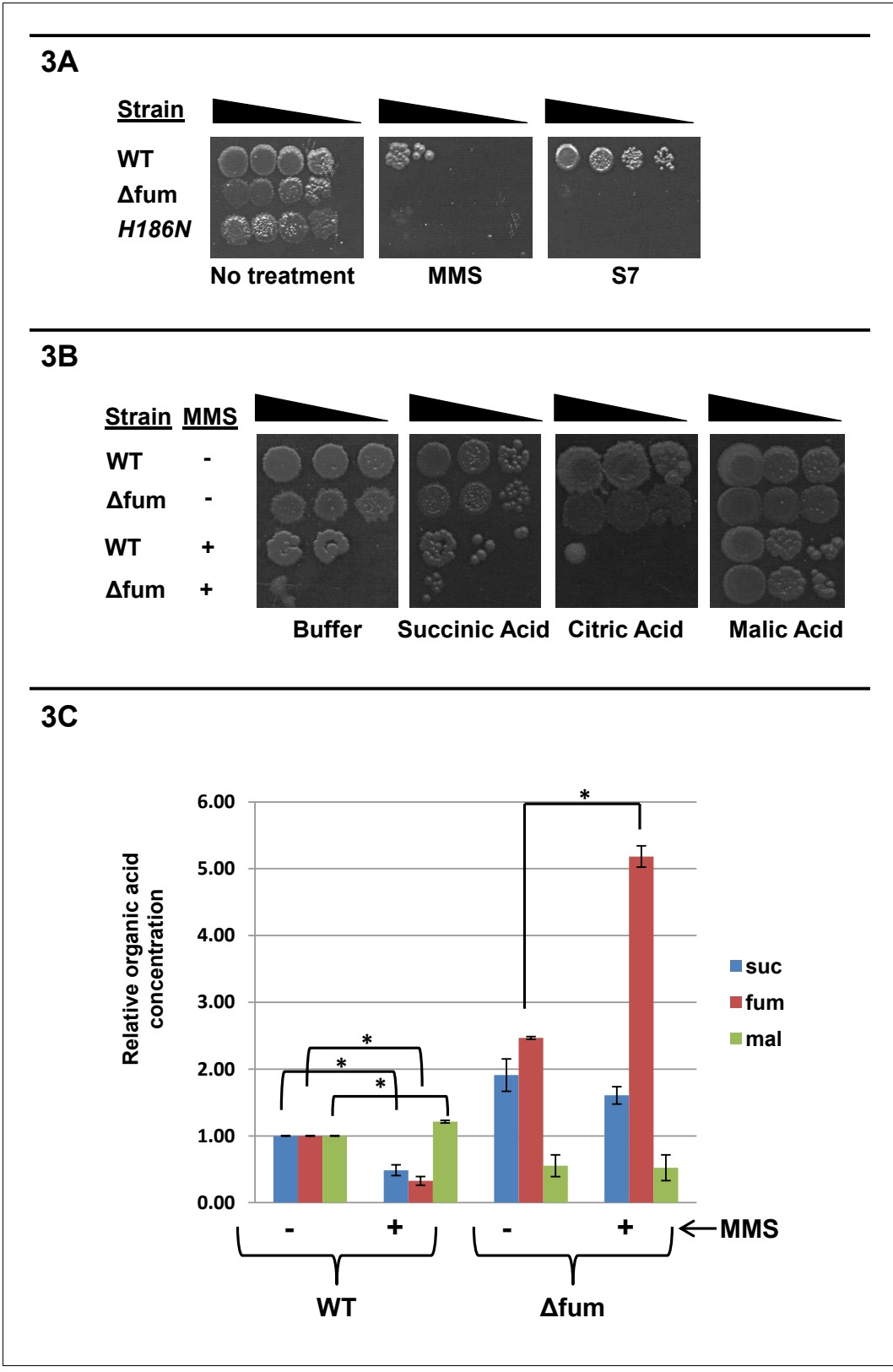

**Figure 3.** Fumarase enzymatic activity and L-malic acid are required for the DNA damage protective function. (**A**) *B. subtilis* wild type, Δfum and mutant H186N fumarase strains were grown to logarithmic phase. MMS was added to a final concentration of 0.07% (v/v) for 45 min, the cells were washed and serially diluted onto LB plates. Untreated cells were serially diluted onto minimal S7 medium plates, containing 0.1% glucose. (**B**) Wild type and
*Figure 3 continued on next page*

*Figure 3 continued*

Δfum strains were grown to logarithmic phase, MMS was added to final concentration of 0.07% (v/v) for 45 min. The cells were washed and serially diluted onto LB plates containing 250 mM of phosphate buffer (pH = 6.8) and the indicated organic acids (Citric acid 10 mM, Malic acid 10 mM, Succinic Acid 25 mM). (**C**) Organic acid accumulation: Wild type and Δfum strains were grown to logarithmic phase, MMS was added to a final concentration of 0.07% (v/v) for 60 min. Metabolite extraction was followed by GC-MS as described in the Materials and methods The graph presents the relative amount of succinic, fumaric or L-malic acid before and after MMS treatment (blue- succinic acid, red- fumaric acid, green- L-malic acid). (mean ±SEM [n = 3], p<0.05). The data presented in *Figure 3A and B*, represent the results of three similar experiments.
DOI: https://doi.org/10.7554/eLife.30927.010

The following figure supplements are available for figure 3:

**Figure supplement 1.** Expression of mutant fumarases.
DOI: https://doi.org/10.7554/eLife.30927.011
**Figure supplement 2.** Enzymatic activity of mutant fumarases.
DOI: https://doi.org/10.7554/eLife.30927.012
**Figure supplement 3.** mutant fumarases.
DOI: https://doi.org/10.7554/eLife.30927.013
**Figure supplement 4.** mutant fumarases.
DOI: https://doi.org/10.7554/eLife.30927.014

L-malic acid by fumarase (succinic - > fumaric - > L malic). Interestingly, upon treatment with MMS the levels of fumaric and succinic acids are even higher (compare the fourth and third sets of bars), indicating an induced flow of metabolites through TCA cycle upon DNA damage.

## Absence of fumarase affects RecN levels and localization

To further implicate *Fum-bc* expression and its function in the DNA damage response, we decided to examine certain *B. subtilis* DDR components in conjunction with fumarase. *RecN* appears to be one of the first proteins recruited to DNA damage sites in live cells (*Cardenas et al., 2014*; *Alonso et al., 2013*). *B. subtilis* cells deleted for the fumarase gene exhibit an alteration in the localization and appearance of RecN (*Figure 4A*, see description below).

*RecN* appears to be the first protein detected as discrete foci (repair centers) in live cells in response to DNA double strand breaks (*Cardenas et al., 2014*). It is cytoplasmically located in untreated cells and upon treatment with DNA damaging agents is recruited to damage sites followed by *RecO* and then *RecF* (*Alonso et al., 2013*). As shown in *Figure 4B,a* strain deleted for RecN shows weak DNA damage sensitivity to MMS (*Sanchez et al., 2007*), while a double knock out of *RecN* and fumarase exhibits an additive effect with the cells exhibiting much higher sensitivity (*Figure 4B*). This effect can be reversed by addition of L-malic acid to the medium (*Figure 4B*, right panel, compare rows 4 and 2 to row 1). Thus, according to the results with *RecN,* we conclude that fumarase is involved in the resistance to DNA damage and its absence can be complemented by L-malic acid.

To examine whether the appearance and localization of RecN-GFP changes upon knockout of the fumarase gene and/or induction of the DNA damage response, we created *B. subtilis* strains harboring genomic RecN-GFP fusions. Following treatment with MMS of wild type cells, there appears to be only a small increase in the proportion of cells containing RecN foci (less than 5%), nevertheless, this increase is statistically significant (*Figure 4—figure supplement 1*), and the appearance of these foci remained unchanged under MMS treatment (*Figure 4A*, compare the RecN-GFP untreated control to RecN-GFP MMS). In strains deleted for the fumarase gene, we observed a similar number of RecN associated foci (*Figure 4A*, third row, RecN-GFP Δfum) when compared to the RecN-GFP control (fourth row). In the presence of MMS (DNA damage) strains deleted for the fumarase gene displayed a drastic change in the number of cells which contain foci (a two fold increase) and in their appearance (*Figure 4A* and *Figure 4—figure supplement 1*, compare RecN-GFP Δfum + MMS to untreated RecN-GFP Δfum). Upon DNA damage the RecN-GFP fluorescence, in the Δfum strain, does not appear as discrete foci but rather this fluorescence coincides with the DNA DAPI stain of the condensed *B. subtilis* chromosomes. Thus, in the absence of fumarase, RecN appears to be recruited differently to the DNA during the DNA damage response.

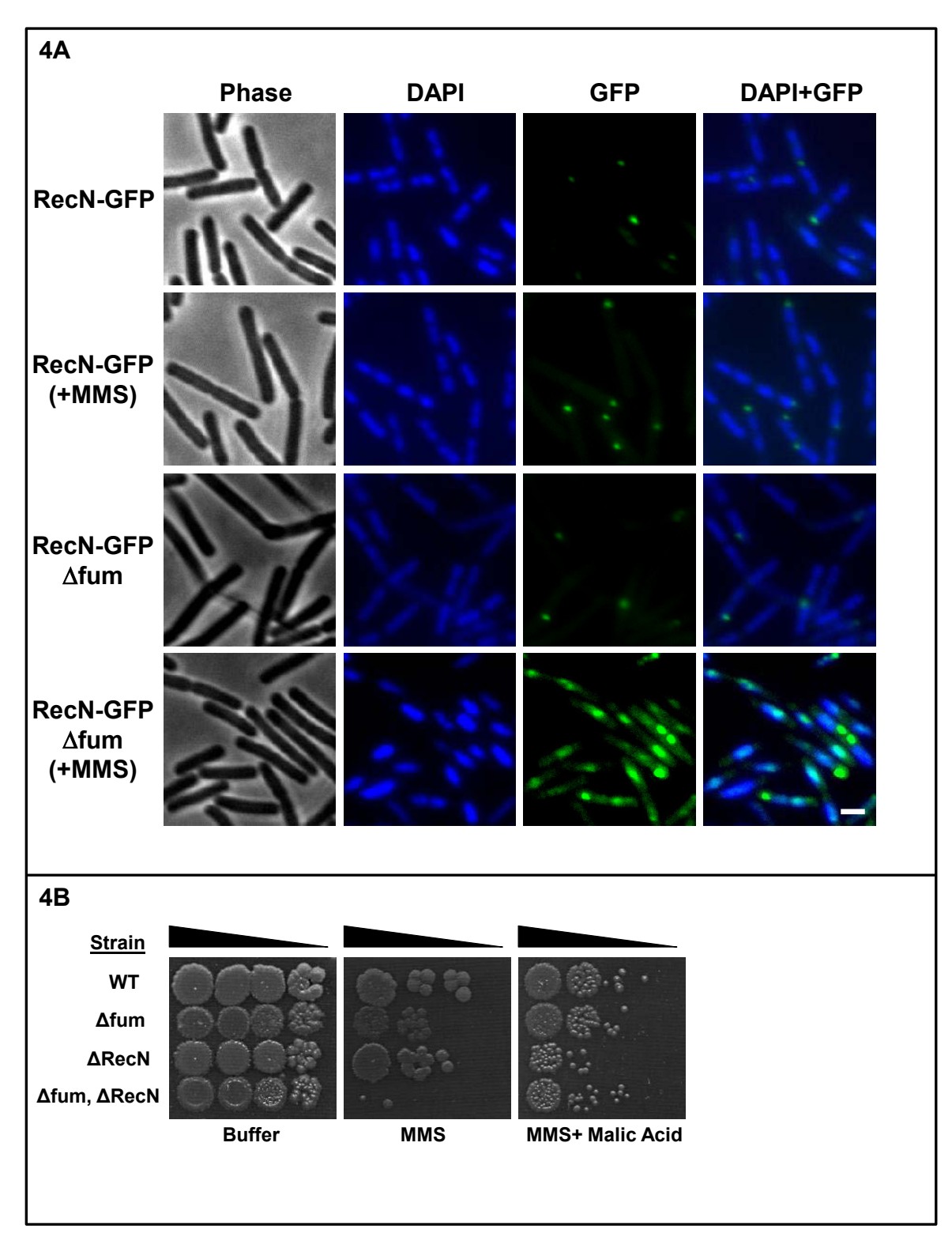

**Figure 4.** Absence of fumarase affects RecN localization. (**A**) Fluorescence microscopy images of *B. subtilis* expressing RecN-GFP in the absence (control) and presence of MMS (upper two rows of panels) and strains lacking the fumarase gene (Δfum, lower two rows of panels). FM4-64 (red), DAPI (blue), GFP (green) and a merge are shown. Representative panels from three independent experiments are shown. Bar, 1 μm. (**B**) Wild type, Δfum, ΔRecN and Δfum/ΔRecN strains were grown to logarithmic phase, MMS was added to a final concentration of 0.07%(v/v) for 45 min. The cells were

*Figure 4 continued on next page*

*Figure 4 continued*
washed and serially diluted onto LB plates containing 250 mM of phosphate buffer (pH = 6.8) and L-malic acid (10 mM). The data represent the results of three similar experiments.
DOI: https://doi.org/10.7554/eLife.30927.015
The following figure supplement is available for figure 4:

**Figure supplement 1.** - Absence of fumarase affects the number of RecN foci.
DOI: https://doi.org/10.7554/eLife.30927.016

In addition to the DNA damage sensitivity and subcellular appearance, we wished to examine the effect of fumarase on RecN levels in the cell. Cells deleted for the fumarase gene were grown to early logarithmic phase (OD = 0.4), MMS was added for 30 min and then cell lysates were subjected to western blot analysis (we could not detect RecN without induction of DSB- data not shown). As shown in *Figure 5A*, RecN is expressed about three fold higher in the strain deleted for the fumarase gene when compared to the wild type (*Figure 5A*, compare lane 2 to lane 1 of the top panel, quantification, *Figure 5B*). In contrast, when L-malic acid is added to the medium the levels of RecN are essentially the same in wild type and strain deleted for the fumarase gene (*Figure 5A*, upper panel, compare the two right lanes, quantitation *Figure 5B*). Worth mentioning is the fact that we have tried to coimmunoprecipitate fumarase and RecN with no positive indications (see the Supplementary methods). Together these data support the notion that fumarase affects RecN function by producing the metabolite L-malic acid and not by direct interaction of the proteins.

The primary goal of this research was to implicate prokaryotic fumarase in the DNA damage response. The finding that fumarase and L-malic acid affect the expression and subcellular appearance of RecN fully supports this hypothesis. Nevertheless, we decided to also examine at which level of gene expression, do fumarase and L-malic acid affect RecN cellular levels. Cells deleted for the fumarase gene and the corresponding wild type were grown to early logarithmic phase (OD = 0.4), MMS was added for 30 min and then the levels of RecN mRNA were determined. We employed quantitative RT-PCR (see materials and methods) and observed no difference between the mRNA levels (data not shown). To further make the point that transcription does not play a role in RecN higher levels in Δ*fum* strains, we grew *B. subtilis* strains in the presence of rifampicin. Rifampicin inhibits bacterial RNA polymerase and blocks transcription initiation, thus, in its presence; changes in protein synthesis do not result from changes in transcription. As shown in *Figure 5C and D*, cells treated with MMS in the presence of rifampicin for 10 or 20 min, revealed significantly higher levels of RecN in the Δ*fum* strain than in the wild type strain. These results indicate that the increased synthesis of RecN, in the absence of fumarase, is due to translation and not transcription. To rule out the possibility that the different RecN levels in strains result from differences in protein stability, we assessed RecN-protein turnover in the presence of chloramphenicol following induction of DNA damage. Chloramphenicol prevents protein chain elongation by inhibiting the peptidyl transferase activity of the bacterial ribosome. As presented in *Figure 5E* the decrease in RecN levels in this experiment shows that both Δ*fum* strain and wild type exhibit similar RecN turnover kinetics (*Figure 5F*).

## Discussion

How did moonlighting and dual targeting of proteins evolve in eukaryotes? The notion is that a protein with a single function, activity or location, acquired new traits through evolution. With regard to fumarase the notion was that the enzyme acquired its second function in the DNA damage response after endosymbiosis and the creation of mitochondria. An example of the acquisition of novel functions after endosymbiosis are roles assumed by yeast Hsp60 and aconitase in mitochondrial genome stability through binding of mitochondrial DNA (*Chen et al., 2005*; *Kaufman et al., 2000*). Fumarase has been shown to distribute between the mitochondria and the cytosol/nucleus by different mechanisms in different eukaryotes(*Yogev et al., 2011*). It seems unlikely that the dual targeting/dual function arose independently in different eukaryotic ancestors. A different possibility is that dual function arose prior to dual localization and actually it was the function that was the driving force for the evolution of fumarase dual targeting.

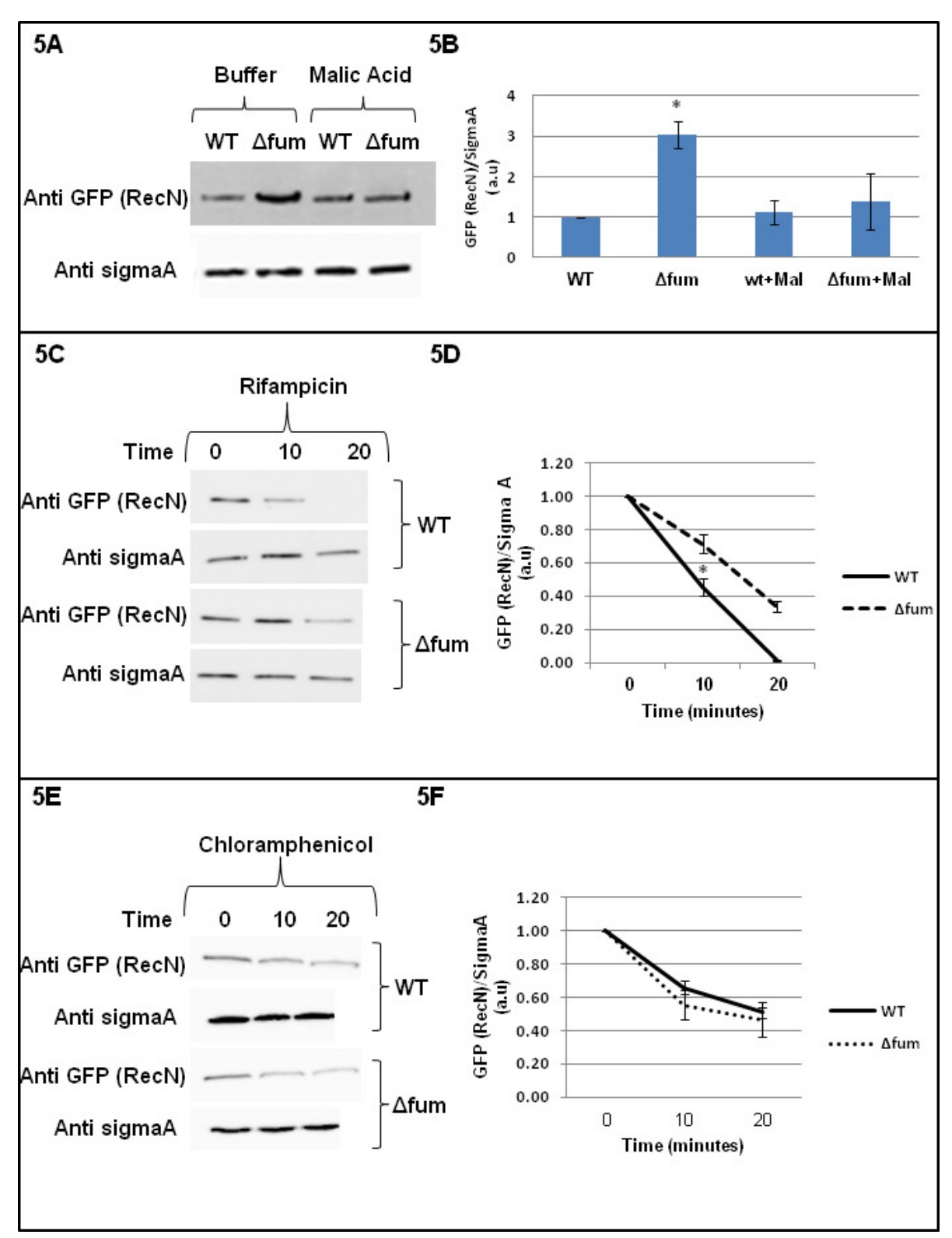

**Figure 5.** Fumarase affects RecN levels. (**A**) RecN level is increased upon DNA damage. Wild type and Δfum strains, expressing RecN-GFP, were grown to logarithmic phase and MMS was added to final concentration of 0.07% (v.v). The cells were plated onto LB plates containing 250 mM of phosphate buffer (pH = 6.8) and L-malic acid, 10 mM. Cells were collected from the plates and the levels of RecN were determined in extracts by western blot analysis. Anti-SigmaA was used as a loading control. (**B**) RecN level is increased upon DNA damage. The chart presents the relative amount of RecN in

*Figure 5 continued on next page*

*Figure 5 continued*

the absence or presence of MMS with or without MMS (densitometric analysis of *Figure 4C* normalized to SigmaA, mean ±SEM [n = 3], p=0.05). (**C**) Wild type and Δfum cells containing RecN-GFP were grown exponentially in LB and exposed to 0.07% (v/v) MMS for 30 min at 37°C. Rifampicin (100 µg/ml) was added to inhibit transcription (time 0). The fate of RecN was followed by western blot and densitometric analysis (normalized to SigmaA and shown as optical density arbitrary units). (**D**) The chart presents the relative amount of RecN after treatment with Rifampicin (densitometric analysis of *Figure 4E* normalized to SigmaA). The RecN concentration at time 0 was arbitrarily determined as 100% and the results represent a mean ±SEM (n = 3), p=0.04. (**E**) Wild type and Δfum cells containing RecN-GFP were grown exponentially in LB and exposed to 0.07% (v/v) MMS for 30 min at 37°C. Chloramphenicol (Cm, 20 µg/ml) was added to inhibit protein synthesis (at time 0). At the times indicated cell growth was halted by addition of NaN3 (10 µM), and the fate of RecN was followed by western blot analysis. (**F**) The chart presents the relative amount of RecN after treatment with chloramphenicol (densitometric analysis of *Figure 5E* normalized to SigmaA). The RecN concentration at time 0 was arbitrarily determined as 100% and the results represent a mean ±SEM (n = 3).
DOI: https://doi.org/10.7554/eLife.30927.017

Our results support a second function for fumarase (in addition to its function in the TCA cycle) in the DDR of bacteria:

- *B. subtilis* deleted for the *citG* gene (*fum-bc*), which encodes fumarase, a TCA cycle enzyme, is sensitive to DNA damaging agents such as ionized radiation (IR) and methyl methanesulfonate (MMS).
- *fum-bc* expressed in a mutant yeast strain can complement the lack of extra-mitochondrial fumarase with respect to sensitivity to DNA damage.
- Upon DNA damage, Fum-bc is induced and is co-localized with the bacterial DNA.
- Fumarase activity and the product of the reaction it catalyzes, L-malic acid, are required for the DDR.
- Absence of fumarase in the bacterial cell affects the levels and localization of the DDR protein RecN.

We have shown that dual-targeted proteins are significantly more evolutionary conserved than exclusive mitochondrial proteins, strongly suggesting that dual function drives the evolution of dual targeting or at least its maintenance (*Ben-Menachem et al., 2011*; *Kisslov et al., 2014*). Our model depicts fumarase as a protein with dual function in the bacterial ancestor of mitochondria. Upon transfer of the endosymbiont gene into the eukaryotic nucleus, the selective pressure due to the two functions that are 'needed to be carried out in different subcellular compartments', resulted in dual targeting of this enzyme. A similar model can be assigned to the dual-targeted yeast aconitase, which is a component of the TCA cycle in mitochondria and the glyoxylate shunt in the cytosol (*Chen et al., 2005*; *Rouault et al., 1991*). This is a simpler example, compared to fumarase, in which aconitase functions in two parallel metabolic pathways that coexist in the prokaryotic cytosol. The main goal of this study was to determine that *B. subtilis* Fum-bc has both TCA cycle and DDR-associated functions. Nevertheless, we have also gained insight into some mechanistic features of fumarase function within the DDR. The most exciting finding is that intracellular signaling of the DDR is achieved via L-malic acid, the product of the reaction catalyzed by fumarase. This conclusion was reached not only due to the fact that L-malic acid, added to the medium, can complement Δ*fum* strains upon DNA damage, but it is also consistent with other data in this study. *B. subtilis* fumarase enzymatic activity is required for its DNA damage protective function. In addition, upon DNA damage *B. subtilis* accumulates L-malic acid versus lower levels of fumaric and succinic acids. Furthermore, upon DNA damage Δ*fum* cells accumulate extremely high levels of fumaric (and succinic) acids as though the cells are 'trying to make more L-malic acid' in order to signal DNA damage. This finding is intriguing since in yeast and human cells, the signaling molecule associated with fumarase, with respect to the DDR, is fumaric not L-malic acid as we find for *B. subtilis*(*Yogev et al., 2011*; *Jiang et al., 2015*). Thus, although the fumarase protein sequence and the dual function/targeting of the enzyme are conserved, the signaling metabolite is different. In other words, when we talk about conservation of function of fumarase in the DDR we do not mean that all aspects of dual function are conserved but rather that the metabolic pathway with specific organic acid intermediates are recruited. This raises questions on how intermediates of primary metabolism were chosen during evolution as signaling molecules in different organisms.

Our results indicate that fumarase does not interact directly with RecN, but rather, the effect is via L-malic acid. How does L-malic acid precisely exert its effect on the DDR? A number of

possibilities come to mind; the organic acid binds components of the DDR directly, thereby, modulating their activity. A good example of such a scenario are succinate and fumarate that can inhibit prolyl hydroxylases (PHDs), resulting in the stabilization of HIF1-α and activation of downstream hypoxic pathways in human cells (*Gottlieb and Tomlinson, 2005*; *Isaacs et al., 2005*; *Pollard et al., 2005*; *Sudarshan et al., 2007*). Another example are local concentrations of fumarate produced by phosphorylated fumarase (bound to histone H2A.Z) which inhibits KDM2B, histone dimethylase, which in turn results in enhancement of histone H3 dimethylation and downstream activation of the DDR (*Jiang et al., 2015*). Another possibility is that L-malic acid affects components of the DDR by affecting their expression. RecN fits the profile of an L-malic acid target since its expression is affected by the acid; the level and appearance of RecN is altered in Δ*fum* cells following DNA damage which is correlated with lower levels of L-malic acid in the cells. RecN does not appear to be regulated at the mRNA level, transcription or protein stability. We claim that L-malic acid affects the translation of RecN mRNA, since upon DNA damage, *B. subtilis* cells lacking fumarase, have three fold higher amounts RecN protein in cell extracts. Importantly, this higher level of RecN can be reversed by growth of the cells in the presence of L-malic acid.

To summarize these notions, the activity of fumarase and L-malic acid are required for an efficient DNA damage response and their effect on RecN has two consequences: (1) A change in the localization of RecN, (upon DNA damage induction), from foci throughout the cell to co-localization with the condensed DNA and (2) A two fold over-expression of RecN at the protein level. While we do not know how lack of fumarase and accumulation of L-malic acid affect the localization of RecN we do know that the change in expression of RecN occurs at the level of translation.

How could L-malic acid affect RecN translation? There are three different ways to temporally regulate gene expression at the translational level: through trans-acting proteins, through cis-acting mRNA elements, acting as riboswitches (*Kirchner and Schneider, 2017*; *Perez-Gonzalez et al., 2016*) and through transacting RNAs (small RNA) (*Kim et al., 2009*). There are no known riboswitches in the RecN gene yet riboswitches have been detected in the upstream gene, *ahrC*, of the RecN operon, which appears to be highly regulated in *Bacillus* (*Dar et al., 2016*). The *ahrC* gene participates in the metabolism of arginine and the riboswitches were identified by term-seq which is quantitative mapping of all exposed RNA 3′ ends in bacteria. This allowed unbiased, genome-wide identification of genes that are regulated by premature transcription termination. Small untranslated RNA SR1, from the *Bacillus subtilis* genome, is a regulatory RNA involved in fine-tuning of arginine catabolism (*Gimpel et al., 2012*). SR1 is an sRNA that acts as a base-pairing regulatory RNA on the *ahrC* mRNA. The interaction of SR1 and *ahrC* mRNA does not lead to degradation of *ahrC* mRNA, but inhibited translation at a post-initiation stage (*Heidrich et al., 2006*). Overexpression of RecN has been shown to be lethal for *B. subtilis* cells. Thus, one of the roles of L-malic acid may be to maintain appropriate RecN levels.

## Materials and methods

### Strains and general methods

*B. subtilis* strains are listed in *Supplementary file 1*, *S. cerevisiae* strains are listed in *Supplementary file 2* of the supplemental material. The plasmids and primers referred to in this study are described in Plasmid construction.

All general methods were carried out as described previously (*Harwood and Cutting, 1990*). Molecular biological methods for Bacillus. Wiley, Chichester, United Kingdom). Cultures were inoculated at an optical density at 600 nm ($OD_{600}$) of 0.05 from an overnight culture, and growth was carried out at 37°C in LB medium. During logarithmic phase ($OD_{600}$ of 0.4 to 0.6), 0.5% xylose or 0.1% IPTG was added to induce citG (fumarase) expression, as indicated.

### Plasmids

**Pfum-GFP** (fum-GFP-kan), containing the 3′ region of fum-bc fused to gfp, was constructed by amplifying the 3′ region by PCR using primers: F̲ GAATTC TTC CAT GAT AAA TGT GCT GT R̲ CTCGAG CGC CTT TGG TTT TAC CAT G, which replaced the stop codon with a XhoI site. The PCR-amplified DNA was digested with EcoRI and XhoI and was cloned into the EcoRI and XhoI sites of pKL168 (*kan*) (*Lemon and Grossman, 1998*), which contains the in frame gfp coding sequence.

Pfum-bc (*amyE::fum-spc*), containing flanking amyE sequences and the spc gene, was constructed by amplifying the citG gene by PCR using primers: F GTCGAC ATG GAA TAC AGA ATT GAA CGA R GCTAGC G CAG CCG TTC TTC CTA TTA. The PCR-amplified DNA was digested with SalI and NheI and cloned into the SalI and NheI sites of pDR111 (amyE::spc).

PH127R, pDR150 [amyE::fum-bc (spec)] is an ectopic integration containing the xylose-inducible promoter. pDR150 was generated by site-directed mutagenesis using the KAPAHiFi™ kit, using primers: F CGT CCA AAT GAT GAC GTG AAC P R A ATC GTT TGA TCA GAG TTC TTC P

PH186N, pDR150 [amyE::fum-bc (spec)] is an ectopic integration containing the xylose-inducible promoter. pDR150 was generated by site-directed mutagenesis using the KAPAHiFi™ kit, using primers: F GAT CTT CAG GAT GCT ACG R CGT GCG TCC GAT TTT GAC

Pyfum yeast expression vector yep51, was constructed by amplifying the citG gene by PCR using primers: F GTCGAC ATG GAA TAC AGA ATT GAA CGA R GGATCC CGC CTT TGG TTT TAC CAT G. The PCR-amplified DNA was digested with SalI and BamHI and cloned into the SalI and BamHI sites of YEp51.

## Fluorescence microscopy

Samples (0.5 mL) of a given culture were removed, centrifuged briefly, and resuspended in 10 µL of PBS × 1 (Phosphate-Buffered Saline) supplemented with 1 µg/mL FM4–64 (Molecular Probes, Invitrogen). Cells were visualized and photographed using an Axioplan2 microscope (Zeiss) equipped with CoolSnap HQ camera (Photometrics, Roper Scientific) or an Axioobserver Z1 microscope (Zeiss) equipped with a CoolSnap HQII camera (Photometrics, Roper Scientific). System control and image processing were performed using MetaMorph 7.2r4 software (Molecular Devices).

## GC-MS analysis

GC-MS analysis of three organic acids was performed using gas chromatograph (Agilent 7890A) coupled to the mass selective (Agilent 5975C MSD). The gas chromatograph was equipped with the CTC COMBI PAL autosampler. Mass spectrometer was operated in SIM mode (single ion monitoring). Plasma samples were dissolved in water following the addition of isotopically labeled succinic acid – D6. The samples were cleaned by SPE (Phenomenex Strata X-AW) and dried over a stream of nitrogen. Acids were chemically derivatized by trimethyl silylation before GC-MS analysis.

## RecN turnover

For RecN half-life determination, strains RecN-GFP and RecN-GFP, Δfum were grown to an OD600 = 0.4 at 37°C in LB. MMS (0.07 v/v) was added to an aliquot, and the cells were incubated for 30 min. Then, rifampicin (100 µg/ml) or chloramphenicol (20 µg/ml) were added. Aliquots were then collected at variable times, and cell growth was halted by addition of NaN3 (10 µM).

## Western blot analysis

Cells were harvested in lysis buffer containing: 10 mM Tris pH 8, 10 mM MgCl$_2$, 0.2 mg/ml AEBSF (MegaPharm-101500), 0.5 mg/ml Lysozyme (USBiological-L9200), 5 µg/ml DnaseI (Sigma DN25). Protein concentrations were determined using Bradford analyses. Samples were separated on 10% SDS-PAGE gels, and then transferred onto PVDF membranes (Millipore). The following primary antibodies were used: Polyclonal anti-yeast fumarase and anti-human FH were generated in rabbits injected with the purified proteins. Monoclonal anti GFP was a product of Roche). Monoclonal anti SigmaA was kindly provided by M. Fujitas lab,. Polyclonal anti ICDH and anti citZ were product of kerafast. Blots were incubated with the appropriate IgG-HRP-conjugated secondary antibody. Protein bands were visualized using the ECL immunoblotting detection system (GE Healthcare) and developed on an ImageQuant LAS4000 mini Fuji luminescence imagining system. For the analysis of protein expression, bands from at least three independent experiments were quantified by densitometry using Image J analysis software.

## RT-PCR

Total RNA was extracted by using a FastRNA blue kit (MP) according to manufacturer's instructions. For integrity assessment and purification level of extracted RNA, 2% agarose gel electrophoresis as well as spectrophotometric assays were performed. The extracted RNA was reverse-transcribed into

cDNA by using a Maxima First Strand cDNA Synthesis Kit with dsDNase (Thermo scientific). To reveal the modification in the expression of RecN, real-time reverse transcriptase PCR (real-time RT PCR) was exploited using SYRB Green dye (Thermo scientific) and the Mic system (Bio Molecular Systems). The raw data were further normalization to the yoxA gene. Three independent experiments were conducted. Sequences of the primers used in the current study were as follows: yoxA: F A TACAATGCGGACGGAAAAC R GGCTCCAGCACTTGTAAACC RecN F CAGGCTCCTTGAACTGC TG R CGT CAG TTC CTC AAT AAT GGC RecN (set2) F TGC ATT ACA CAC CTG CCT CA R CGC TAC CTT TTC CTG CTT AG

## Statistical analysis

When more than two groups were compared, statistical analysis was performed by one-way repeated measure analysis of variance with Duncan's test. When only two groups were compared, significance was analyzed by the paired t test.

# Additional information

## Funding

| Funder | Author |
| --- | --- |
| Israel Science Foundation | Ophry Pines |
| German Israeli Project Cooperation | Ophry Pines |
| CREATE Project of the National Research Foundation of Singapore | Ophry Pines |

The funders had no role in study design, data collection and interpretation, or the decision to submit the work for publication.

## Author contributions

Esti Singer, Resources, Validation, Investigation, Visualization, Methodology, Writing—original draft, Writing—review and editing; Yardena BH Silas, Validation, Investigation, Visualization, Methodology, Writing—review and editing; Sigal Ben-Yehuda, Methodology, Writing—review and editing; Ophry Pines, Conceptualization, Resources, Supervision, Funding acquisition, Validation, Writing—original draft, Project administration, Writing—review and editing

## Author ORCIDs

Ophry Pines http://orcid.org/0000-0001-7126-2575

## Decision letter and Author response

Decision letter https://doi.org/10.7554/eLife.30927.023
Author response https://doi.org/10.7554/eLife.30927.024

# Additional files

## Supplementary files

• Supplementary file 1. *B. subtilis* strains. *B. subtilis* strains used in this study and their source.
DOI: https://doi.org/10.7554/eLife.30927.018
• Supplementary file 2. *S. cerevisiae* strains. *S. cerevisiae* strains used in this study and their source.
DOI: https://doi.org/10.7554/eLife.30927.019
• Transparent reporting form
DOI: https://doi.org/10.7554/eLife.30927.020

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
