## [Decision Letter]

Thank you for submitting your article "Bacterial fumarase and L-malic acid are evolutionary ancient components of the DNA damage response" for consideration by *eLife*. Your article has been reviewed by three peer reviewers, and the evaluation has been overseen by Klaus Pfanner as the Reviewing Editor and Randy Schekman as the Senior Editor. The following individual involved in review of your submission has agreed to reveal his identity: Doron Rapaport (Reviewer #1).

The reviewers have discussed the reviews with one another and the Reviewing Editor has drafted this decision to help you prepare a revised submission.

Summary:

Singer and colleagues present data suggesting that dual functions of a single protein evolved prior dual localization. Fumarase is an enzyme found in both the cytoplasm and mitochondria of all eukaryotes. In mitochondria, it is involved in the Krebs cycle whereas the cytoplasmic version of fumarase has a role in repairing DNA double-strand breaks in the nucleus. This role involves the movement of fumarase from the cytoplasm into the nucleus and depends on its enzymatic activity. In the present contribution, the authors nicely demonstrate that such dual function occurs already in prokaryotes like *B. subtilis*. They provide convincing evidence that fumarase catalytic activity is required to mediate the DNA damage response in prokaryotic cells. As such, mutants lacking Fum-bs are sensitive to DNA damage stress and fail to recruit important proteins needed to repair damaged DNA. Furthermore, these phenotypes can be rescued by adding exogenous malate to growth media. In the end, the evidence presented quite clearly supports the hypothesis that the dual function of fumarase in the DNA damage response and the TCA cycle evolved in prokaryotes prior to the divergence of eukaryotes. Thus, the authors propose that proteins with dual functions in prokaryotes drove the evolution of dual localization once compartments such as mitochondria and nuclei came to exist.

In general, the manuscript is well organized with high quality data that support their hypothesis.

Essential revisions:

1) Does Fum-bc contain a mitochondrial presequence? The authors should test whether Fum-bc (or rather Fum-bc-GFP to allow specific immunodetection) is imported into the mitochondrial matrix or was present only in the cytosol. Testing import can be performed also by an in vitro import assay. According to the results, they should discuss how Fum-bc, which is maybe only in the cytosol, can complement the TCA cycle defect in fumΔ yeast cells.

Figure 1. It is unclear how much Fum-bc is imported into mitochondria. The authors should verify that Fum-bc is efficiently imported into mitochondria.

In yeast, Fum1 is retro-translocated from mito to cytosol. Is FUM1m exclusively mitochondrial or is it retro-translocated to even a small extent?

2) Are the cells larger following MMS treatment? An enlarged image demonstrating that in the absence of MMS, Fumarase-GFP and DAPI do not overlay is important.

The authors should exclude the possibility that treatment with MMS resulted in higher expression of many TCA enzymes (and not only Fum-bc). A possibility that is supported by their conclusion at the end of subsection “Fumarase activity and the product of its reaction, L-malic acid, are required for the DDR”. A Western blotting with additional TCA enzyme(s), as a control, is required.

Figure 2. 2A shows that Fum-bc accumulates upon MMS treatment. This increase in expression is not reflected in the subsequent microscopy. Please explain this discrepancy. If the microscopy is differentially exposed between strains, this should probably be corrected. If two different exposures need to be shown to fully capture the effect, that's fine.

3) The authors nicely show the requirement of fumarase and malic acid for growth under DNA damaging conditions. However, Figure 3 demonstrates that MMS causes increased levels of malic acid, whereas Figure 5 suggests that malic acid (as product of fumarae) rather inhibits the expression of RecN. How can these observations come together? It will help the manuscript if the authors can show an indication for the involvement of malic acid in the actual DNA repair mechanism.

Figure 4. Malate can rescue the growth defects in the presence of MMS. In order to connect the growth phenotype with the RecN phenotype, the authors should perform the microscopy experiments with malate added to the media.

Figure 4. The authors claim that RecN mutants are sensitive to MMS. This is an important point but the data presented are not at all convincing. Please provide an explanation or more compelling data.

4) A concern is that the author's data suggest that dual localization (in addition to dual function) already existed in prokaryotes (Figure 2). Although, it is unclear what the non-genomic "compartment" may be. A more thorough description for genomic DNA and Fumarase-GFP overlay would be appreciated in the experiment in Figure 2. As is, it is a bit confusing. Without MMS, the DAPI stain appears to fill the entire cytosol, with several Fumarase-GFP spots throughout that appear to at least partially co-localize. Upon MMS treatment, the genome appears to condense into a single spot, as does Fumarase-GFP which now very clearly co-localizes. Thus, their hypothesis that dual function preceded dual localization may be true, but the dual localization in prokaryotes should at least be considered throughout the manuscript.

5) The following point should be addressed at least in a detailed discussion. If prokaryotes use malate as their connecting metabolite and eukaryotes use fumarate, then this calls into question whether this is conservation. What is the role of fumarase in the DNA damage response? It remains unclear how fumarase effects RecN recruitment and whether this is a direct result of malate. It is curious that prokaryotes and eukaryotes use a different metabolite to support DNA damage response and the authors should address this discrepancy.

6) The following technical points should be addressed:

Figure 1. Rescue the phenotypes presented with a plasmid.

Figure 1. FUM1m growth assay should be conducted on ethanol medium.

Figure 3. Include a WT Fum-bc to the growth assay.

Figure 5. The authors need to measure the levels of mRNA in these cells.

7) The text of the paper could use some added clarity.

---

## [Author Response]

Essential revisions:

*1) Does Fum-bc contain a mitochondrial presequence? The authors should test whether Fum-bc (or rather Fum-bc-GFP to allow specific immunodetection) is imported into the mitochondrial matrix or was present only in the cytosol. Testing import can be performed also by an* in vitro *import assay. According to the results, they should discuss how Fum-bc, which is maybe only in the cytosol, can complement the TCA cycle defect in fumΔ yeast cells.*

Figure 1. It is unclear how much Fum-bc is imported into mitochondria. The authors should verify that Fum-bc is efficiently imported into mitochondria.In yeast, Fum1 is retro-translocated from mito to cytosol. Is FUM1m exclusively mitochondrial or is it retro-translocated to even a small extent?

Fum-bc does not contain a mitochondrial targeting sequence, and therefore it is not targeted to or imported into mitochondria. To address the reviewer's concern, we have now performed subcellular fractionation of yeast cells expressing fum-bc, and as expected the bacterial protein is located only in the "cytosol" and not in mitochondria (Figure 1—figure supplement 3). Previous studies from our group indicate that fumarase, located outside mitochondria, can at least partially function in the TCA cycle. The explanation for this is that the metabolites fumarate and malate can enter and exit the organelle (via specific inner membrane transporters), thereby completing the TCA cycle. All this is also valid for yeast Fum1 lacking a targeting sequence (18,19) and *E. coli* fumC (unpublished data). We have now added text and a supplementary figure to make this point clear. (Figure 1—figure supplement 3, subsection “Bacterial fum-bc functions in the TCA cycle and the DDR”).

Yeast fumarase in the FUMm strain is transcribed from the mitochondrial genome translated on mitochondrial ribosomes and is exclusively mitochondrial. The text of the manuscript clearly refers to this point (subsection “Bacterial fum-bc functions in the TCA cycle and the DDR”, Yogev et al., 2010).

2) Are the cells larger following MMS treatment? An enlarged image demonstrating that in the absence of MMS, Fumarase-GFP and DAPI do not overlay is important.The authors should exclude the possibility that treatment with MMS resulted in higher expression of many TCA enzymes (and not only Fum-bc). A possibility that is supported by their conclusion at the end of subsection “Fumarase activity and the product of its reaction, L-malic acid, are required for the DDR”. A Western blotting with additional TCA enzyme(s), as a control, is required.Figure 2. 2A shows that Fum-bc accumulates upon MMS treatment. This increase in expression is not reflected in the subsequent microscopy. Please explain this discrepancy. If the microscopy is differentially exposed between strains, this should probably be corrected. If two different exposures need to be shown to fully capture the effect, that's fine.

*B. subtilis* cells treated with MMS are not larger, yet display a change in morphology mainly in the condensation of the nucleoid and the cells appear as elongated rods due to inhibition of cell division. Upon DNA damaging treatment, fumarase clearly coincides with the condensed DNA. Without DNA damaging treatment the fum-bc-GFP foci and DAPI stained DNA do not generally superimpose. Nevertheless, one does see some co-staining, which can be explained by simple coincidence or naturally occurring random DNA damages. We quantified this by determining the proportion of fum-bc-GFP foci which coincide with the DNA DAPI stain (Figure 2). Please see below a figure, added for the reviewers, in which one can see that upon MMS treatment, undivided bacilli rods and partially condensed DNA coincide with fumarase-GFP.

It is important to note that we do not measure the level of total fluorescence but rather the number of foci per cell. Another point worth making is that the level of fumarase increases significantly upon DNA damage which is reflected in western blots (Figure 2).

Upon induction of DNA damage the level of Fum-bc increases significantly (Figure 2). As requested by the reviewers, we checked other enzymes of the TCA cycle for we could obtain antibodies. As now shown In Figure 2—figure supplement 1, the levels of citrate synthase 2 (citZ) and icocitrate dehydrogenase (ICDH) do not change significantly upon treatment with MMS (Figure 2—figure supplement 1, subsection “Fum-bc is induced upon DNA damage, and is co-localized with the bacterial DNA”).

3) The authors nicely show the requirement of fumarase and malic acid for growth under DNA damaging conditions. However, Figure 3 demonstrates that MMS causes increased levels of malic acid, whereas Figure 5 suggests that malic acid (as product of fumarae) rather inhibits the expression of RecN. How can these observations come together? It will help the manuscript if the authors can show an indication for the involvement of malic acid in the actual DNA repair mechanism.

This is an important point raised by the reviewer, which we now accordingly discuss in the manuscript (Discussion section). The activity of fumarase and L-malic acid are required for an efficient DNA damage response, and their effect on RecN has two consequences: 1) A change in the localization of RecN, (upon DNA damage induction), from foci throughout the cell to co-localization with the condensed DNA, and 2) A two-fold over-expression of RecN at the protein level. We know that the lack of fumarase and accumulation of L-malic acid brings about a change in expression of RecN at the level of translation. This also has an etiological explanation; overexpression of RecN has been shown to be lethal for *B. subtilis* cells. Thus, one of the roles of L-malic acid may be to maintain appropriate RecN levels. With that being said, we do not know with whom L-malic acid interacts in order to affect RecN localization. Attempts to show a direct interaction between L-malic acid and RecN have been unsuccessful, neither supporting nor ruling out such an interaction.

Figure 4. Malate can rescue the growth defects in the presence of MMS. In order to connect the growth phenotype with the RecN phenotype, the authors should perform the microscopy experiments with malate added to the media.

Attempts to perform microscopy on bacterial cells grown in the presence of organic acids such as L-malic acid, succinic acid or citric acid have been unsuccessful; the organic acids at high concentration inhibit growth and the morphology of the cells that were treated with the different organic acids appear abnormal. Growth on agar plates on the other hand, as presented in the manuscript (Figure 5), is fine, yet these cells are unsuitable for microscopy experiments.

Figure 4. The authors claim that RecN mutants are sensitive to MMS. This is an important point but the data presented are not at all convincing. Please provide an explanation or more compelling data.

As reported in published studies (Kokko et al., 2006), although RecN is the first factor recruited to the site of DNA damage, *B. subtilis* deleted for this gene exhibits weak DNA damage sensitivity to MMS (Sanchez et al., 2007), and this is what we find in our current study. (subsection “Absence of fumarase affects RecN levels and localization”).

4) A concern is that the author's data suggest that dual localization (in addition to dual function) already existed in prokaryotes (Figure 2). Although, it is unclear what the non-genomic "compartment" may be. A more thorough description for genomic DNA and Fumarase-GFP overlay would be appreciated in the experiment in Figure 2. As is, it is a bit confusing. Without MMS, the DAPI stain appears to fill the entire cytosol, with several Fumarase-GFP spots throughout that appear to at least partially co-localize. Upon MMS treatment, the genome appears to condense into a single spot, as does Fumarase-GFP which now very clearly co-localizes. Thus, their hypothesis that dual function preceded dual localization may be true, but the dual localization in prokaryotes should at least be considered throughout the manuscript.

The reviewer refers to the main point of the manuscript: In *B. subtilis*, there are no internal membranes, which delimit subcellular compartments. Thus, in the prokaryote, localization of fumarase to the complexes of the TCA cycle, and to complexes of the DDR (on the DNA) may be considered the ancient precursor to dual targeting of the enzyme to two “bone fide” subcellular compartments of the eukaryotic cell. Precisely repeating point 2 above: "Upon DNA damaging treatment, fumarase clearly coincides with the condensed DNA. Without DNA damaging treatment the fum-bc-GFP foci and DAPI stained DNA do not regularly superimpose. Nevertheless, one does see some co-staining which can be explained by simple coincidence or naturally occurring low random DNA damage for example at DNA replication sites. We quantified this by determining the proportion of fum-bc-GFP foci which coincide with the DNA DAPI stain (Figure 2)." (subsection “Fum-bc is induced upon DNA damage, and is co-localized with the bacterial DNA”).

5) The following point should be addressed at least in a detailed discussion. If prokaryotes use malate as their connecting metabolite and eukaryotes use fumarate, then this calls into question whether this is conservation. What is the role of fumarase in the DNA damage response? It remains unclear how fumarase effects RecN recruitment and whether this is a direct result of malate. It is curious that prokaryotes and eukaryotes use a different metabolite to support DNA damage response and the authors should address this discrepancy.

We fully understand the question regarding the finding that L-malic acid in *B. subtilis* versus fumaric acid in eukaryotes are employed as signaling molecules of the DDR. When we discuss conservation of function of fumarase in the DDR, we do not mean that all aspects of dual function are conserved but rather that the metabolic pathway with specific organic acid intermediates are recruited. We refer to this point in the manuscript discussion(Discussion section). To make the point here we refer to another manuscript of ours that will be submitted shortly, showing that in yeast fumarase and fumarate, signal the homologous recombination (HR) DNA repair pathway, whereas, in human cells the same enzyme and metabolite signal the non-homologous end joining (NHEJ) DNA repair pathway. It is important to point out that although the same signaling metabolite is employed (fumarate) the factors it affects downstream are different. Worth mentioning is that fumarase associates with different protein partners in the two compartments (in the mitochondria where it part of the TCA cycle it interacts with other TCA cycle enzymes such as malate dehydrogenase, while in the cytosol/nucleus it interacts with kinases and histones which are part of the DDR).

6) The following technical points should be addressed:Figure 1. Rescue the phenotypes presented with a plasmid.

The figure has been replaced with a figure containing the control requested by the reviewer, Figure 1

Figure 1. FUM1m growth assay should be conducted on ethanol medium.

FUM1m grows less efficiently on ethanol plates so for DDR related experiments we use galactose. We have previously used ethanol medium to show that FUM1m strains can grow on two-carbon energy source media (Yogev et al., 2010), which require the TCA cycle and respiration.

Figure 3. Include a WT Fum-bc to the growth assay.

The first row of Figure 3 shows the WT Fum-bc. We used the WT cells because we do not see any difference in growth on agar plates between the WT cells and the Δfum+ Fum-bc cells with regard to the DDR and respiration. As shown in Figure 1 the WT strain and the Δfum+ Fum-bc strain exhibit similar growth after induction of DNA damage or following growth on low glucose media (S7). In addition, in figure Figure 1—figure supplement 2 the enzymatic activity assay of cell lysates in-vitro shows the two strains are the same.

Figure 5. The authors need to measure the levels of mRNA in these cells.

RecN mRNA levels were determined. We employed quantitative Real Time PCR and observed no difference between the mRNA levels (subsection “Absence of fumarase affects RecN levels and localization”)

7) The text of the paper could use some added clarity.